# Design of Physicochemical Properties of Eggs as a Result of Modification of the Fat Fraction of Laying Feed

**DOI:** 10.3390/molecules29061242

**Published:** 2024-03-11

**Authors:** Agnieszka Filipiak-Florkiewicz, Maja Dymińska-Czyż, Beata Szymczyk, Magdalena Franczyk-Żarów, Renata Kostogrys, Adam Florkiewicz, Marcin Lukasiewicz

**Affiliations:** 1Faculty of Food Technology, University of Agriculture in Krakow, al. Mickiewicza 21, 31-120 Krakow, Poland; agnieszka.filipak-florkiewicz@urk.edu.pl (A.F.-F.); maja.dyminska-czyz@student.urk.edu.pl (M.D.-C.); magdalena.franczyk-zarow@urk.edu.pl (M.F.-Ż.); renata.kostogrys@urk.edu.pl (R.K.); adam.florkiewicz@urk.edu.pl (A.F.); 2National Research Institute of Animal Production, ul. Krakowska 1, 32-083 Balice, Poland; beata.szymczyk@iz.edu.pl

**Keywords:** functional food, eggs, unsaturated fatty acids

## Abstract

The aim of this study was to investigate and confirm the properties of eggs produced by laying hens fed a diet consisting of pomegranate seed oil as a source of CLnA and flaxseed oil as a source of α-linolenic acid. The study involved determining the chemical composition of the eggs, including their fatty acid profile. The results showed that modifying the laying hens’ feed composition resulted in eggs with high nutritional value, with a favorable change in their fatty acid profile. In most cases, the addition of linseed oil or pomegranate seed oil did not affect the physical and chemical properties of the eggs. However, the diet of laying hens had a positive effect on the fatty acid profile of the egg yolk. The presence of conjugated linolenic acid trienes in eggs produced by laying hens fed a modified diet makes them a potential source of these compounds for human consumption.

## 1. Introduction

Today, there is no doubt that the type and quality of food consumed have a strong impact on human health. As a result, many studies are focused on improving food products, with a significant amount of research devoted to modifying the composition and properties of eggs [1].

Contemporary methods of nutritional modification of the composition of hen eggs mainly concern the lipids of the yolk, including the cholesterol content and the composition of fatty acids through the selection of layers or nutritional and pharmacological methods [2]. Nutritional modifications may also encompass the mineral composition and vitamin content [3,4,5]. The fatty acid profile of the yolk depends primarily on its composition in the feed, which allows easy modification, especially towards the proportion of mono- and poly-unsaturated fatty acids [6,7].

One method of improving egg lipid composition through dietary modification involves feeding laying hens a diet rich in olive oil, particularly the C18:1 fatty acid (oleic acid). This results in a linear increase in the concentration of oleic acid in egg yolk lipids [4]. However, the greatest interest is raised by the possibility of modifying the proportion of polyunsaturated fatty acids in egg yolks and the *n*-6 and *n*-3 fatty acid ratios [8,9,10,11]. The methods of enriching eggs with polyunsaturated fatty acids from the *n*-3 family include feeding hens a diet containing linseed or linseed oil [12,13]. Modification allows the production of eggs with higher α-linolenic acid and DHA (docosahexaenoic acid) contents [14]. Another method is to add fish oils to the hen feed. However, this results in a change in the sensory characteristics of eggs (noticeable fishy taste in the yolk) [12]. At the same time, it is known from the literature data that polyunsaturated fatty acids are easily susceptible to oxidation [15]. This disadvantage can be minimized by the addition of antioxidant substances that inhibit the unfavorable transformation of these acids [16,17,18].

In contrast, feeding α-linolenic acid to laying hens resulted in its direct incorporation into the lipid composition of the yolk, but with a significantly lower increase in the proportion of α-linolenic acid in the yolk compared to the dietary concentration in the feed [19].

Enrichment of eggs with *n*-3 fatty acids has attracted the attention of both scientists and the food industry because these acids are essential for the normal development of the human body and play an important role in the prevention of heart disease, diabetes, inflammation, autoimmune diseases, and cancer [20,21]. These ingredients have a positive effect on the blood supply to the brain and improve the flow of nerve stimuli between the gray and white matter. A daily diet enriched with eggs, which is designed, functional, and has enhanced health-promoting properties, can be beneficial to the body [22,23].

Many studies have also focused on the possibility of enriching eggs with conjugated linoleic acid dienes (CLA), which occur naturally in foods of animal origin [24,25]. With an increase in the content of these isomers in the feed of laying hens, an increase in their levels in the lipids of egg yolk was observed. In addition, the content of saturated fatty acids (myristic, palmitic, and stearic acids) increases, while the levels of monounsaturated (oleic) and polyunsaturated (linoleic, linolenic, arachidonic, and DHA) acids decrease [24,25,26]. For many years, numerous studies have been conducted on the health-promoting properties of CLA [27].

Recent studies have discovered that in addition to conjugated linoleic acid, there is a group of compounds called conjugated linolenic acid (CLnA) isomers that may have beneficial effects on the human body. These are polyunsaturated fatty acids with at least three double bonds in their structures [28]. They include α-eleostearic acid (cis-9, trans-11, trans-13) and punicic acid (cis-9, trans-11, cis-13) [29]. The CLnA isomers have the potential to exhibit high biological activity depending on their structure [30]. These compounds are naturally present in large amounts in certain oils, such as cucumber tung oil, pomegranate oil, catalpa oil, and karela oil. Depending on the type of oil, different CLnA isomers can be observed [30,31]. Research results indicate that CLnA isomers may have health-promoting properties, including contributing to the reduction of adipose tissue and having anti-cancer properties [31,32]. Tsuzuki et al. also demonstrated that CLnA isomers can be transformed into conjugated linoleic acid dienes in rats [33].

One of the richest sources of CLnA is the pomegranate fruit (*Punica granatum* L.), a plant belonging to the *Punicaceae* family [32]. The edible part of the pomegranate fruit constitutes approximately 55–60% of its total weight and consists of approximately 75–85% of the juice and 15–25% of the seeds [34,35]. Pomegranate seed oil has high CLnA content (31–86%) and contains linoleic (1–24%), oleic (0.5–17%), stearic (16–23%), and palmitic (0.3–10%) acids. The differences in the fatty acid profiles result from varietal characteristics [36]. However, when it comes to its functional properties as a source of conjugated linolenic acid trienes, studies indicate a possible inhibitory effect of CLnA on cancer cell growth, as well as a lowering of liver cholesterol levels [37]. Additionally, pomegranate seed oil has been found to reduce triacylglycerol levels in the liver under the influence of punicic acid, its main component [38]. Moreover, pomegranate seed oil, which is primarily composed of punicic acid, possesses strong anti-inflammatory and antioxidant properties [39]. Its high radical scavenging activity as a result of the presence of natural antioxidants including tocopherols makes it an excellent source of these beneficial compounds [40,41].

Research suggests that conjugated linolenic acid trienes may be beneficial in preventing and treating various diseases, including diabetes, atherosclerosis, obesity, and various types of cancer (such as breast, prostate, and colon cancer). Pomegranate seed oil has been found to improve the stability of atherosclerotic plaques in mice when added to the diet of laying hens in the amount providing 1.5% CLnA [4]. However, the high cost of these eggs raises questions about their practical application. In this study, an attempt was made to produce eggs based on feeding laying hens with feed containing pomegranate seed oil and linseed oil as sources of CLnA and α-linolenic acid, respectively.

According to that, the following research hypothesis was formulated in this study: as a result of modifying the feed composition of laying hens, the eggs will retain their organoleptic properties and high nutritional value while favorably changing the fatty acid profile.

## 2. Results and Discussion

The initial stage of this investigation involved the determination of the physicochemical properties of fresh eggs. The weight of the analyzed eggs was found to range from 60.46 g to 63.14 g, with minimal variations observed based on the composition of the laying hen feed. The values of the egg shape index and Haugh units were typical for fresh eggs and showed no significant discrepancies among groups in the first as well as second experiments, as presented in Table 1. Furthermore, the percentages of shell, egg white, and yolk in the tested material did not exhibit any significant variations based on the type and quantity of oil incorporated into the hen feed. Similarly, when evaluating yolk color using the 15-point La Roche scale, no significant differences were observed.

No notable alterations were also detected in the chemical composition of eggs with regard to their total fat content and ash content in egg whites (as presented in Table 2). Similar results by means of fat content were obtained in another study [42], where flaxseed oil was added to the feed.

### 2.1. Experiment 1

With an augmentation in the quantity of pomegranate seed oil (PSO) in laying hen feed, the pH values of egg whites increased statistically significantly, while those of egg yolks remained unchanged (*p* > 0.05) in all experimental cases. Moreover, the egg white index of egg D was found to be significantly lower than the control eggs. No substantial differences in yolk index values were observed in the experiment where hens were fed linseed oil (LO) in the feed. However, the instrumental measurement of color revealed that it varied in accordance with the feed additive utilized. Egg yolk D exhibited a significantly lower total color difference (ΔE = 2.75) compared to the other samples. The alterations in the ΔE were accompanied by changes in L, a, and b parameters as well.

Table 2 presents the basic chemical composition of the eggs. The dry matter content in the egg yolk fluctuated between 47.27 g/100 g and 48.08 g/100 g and was contingent upon the level of PSO in the diet of laying hens. Moreover, the dry matter content in egg white was 13.44 g/100 g in the control group A and declined in the other groups (B–D). It should be noted that egg whites from groups fed 1.0% PSO (C) and 1.5% PSO (D) may be characterized by the lowest dry matter content. Similarly, Siepka et al. [43] discovered a correlation between the enrichment of hen feed with *n*-3 acids, contrasting the results of Piva et al.’s research on the effects of *n*-3 acids on dry matter content in egg whites [44].

The level of protein in the dry matter of egg whites increased with the increasing proportion of PSO in the hen’s feed. No significant differences were observed in the protein content of the egg yolk. The egg yolks’ fat content varied from 49.18 g/100 g (group C) to 52.99 g/100 g (group D). Additionally, there were no significant differences in the ash content of the egg whites.

Based on Table 3, it may be stated that no statistically significant changes were observed in the levels of saturated fats (*p* > 0.05) in hens fed with the increase of PSO. However, there were variations in the levels of individual acids in some cases. For instance, significant changes were observed for C14 acid, which showed the highest level of acid in group C. This indicates the impact of the use of PSO in this variant. On the other hand, a decrease in C16 acid content was observed in group C, accompanied by an increase in the level of stearic acid (C18) in egg yolks when a higher amount of PSO was used in the hen feed.

The control egg yolk A had a slightly higher percentage of monounsaturated acids than the remaining eggs (*p* > 0.05), at 50.02%; however, no significant decrease in the proportion of monounsaturated fatty acids in egg yolk lipids was noted (*p* > 0.05). For the individual MUFAs, only trans-3-hexadecenoic acid and oleic acid exhibited alterations in their content. Specifically, the highest concentrations of trans-3-hexadecenoic acid were observed in group A (feed with no PSO in feed). Variant A showed also the highest concentrations of oleic acid.

A significant increase in the proportion of both CLnA and CLA in the fatty acid profile was found, dependent on the amount of PSO in the feed. The higher the amount of PSO, the greater the proportion of these acids in the yolk lipids. An increase in the amount of arachidonic acid (C20:4 *n*-6) and docosahexaenoic acid (C22:6 *n*-3) was also observed, while no significant changes in the case of the other investigated C20 isomers were detected.

It is also worth pointing out that in control group A, the dry matter content in the egg white was 13.44 g/100 g, which decreased as the level of conjugated linolenic acid trienes in the feed increased. Piva et al. obtained similar [44], whereas Dziadek et al. observed lower values [45].

### 2.2. Experiment 2

In Experiment 2, eggs obtained from hens fed a diet containing 1.5% PSO (H) exhibited a significantly lower yolk index compared to the other eggs (Table 1). Different relationships were observed for the egg white index, and it was observed that group E had a higher value. Furthermore, it was observed that as the proportion of PSO in the hens’ feed increased, the color difference between the yolks of PSO-supplemented eggs and control eggs also increased, with a maximum value of ΔE = 11.23 observed for egg yolks in group H.

The highest lightness was recorded for the egg yolks in group E. In Experiment 2, the same trend was observed for the blue/yellow values; however, the changes observed were not statistically significant.

The dry matter content of egg yolks in the study varied between 46.11 g/100 g and 47.78 g/100 g and was found to be dependent on the level of PSO in the diet for laying hens, as observed in Experiment 1. The dry matter content of egg white varied with increasing PSO content in the diet, with the maximum value in group E and minimum in F. Examination of the protein content of egg yolks revealed a decrease with increasing levels of PSO added to the feed of laying hens. Specifically, a significantly lower protein content was observed in egg yolks from group H (from hens fed 1.5% PSO) than in the other yolks in Experiment 2. Lower values of that parameter were defined by Siepka et al. for egg yolks enriched with *n*-3 acids [43]. A different pattern was discerned in the protein content of the egg whites in this study. Specifically, an increasing trend in protein content was observed as the content of PSO in the feed increased. The fat content of egg yolks varied from 52.24 g/100 g (group E) to 49.92 g/100 g (group H) and did not significantly differ depending on the type of oil in the feed of laying hens. The results go in line with that obtained in other research that used flaxseed in hen’s feed [42]. However, variability in the ash content of egg yolks was observed in Experiment 2, with eggs from all experimental groups (F–H) containing less ash than the control group (E).

Undertaking an analysis of the findings from Experiment 2 with respect to the saturated acid content in egg yolks (Table 3), it can be reported that no discernible variations in the concentration of this compound were observed in response to alterations in the PSO level in the feed. Some modifications were observed in the distribution of individual compounds within this group; specifically, an increase in PSO levels in the feed was accompanied by an increase in the quantity of stearic acid, whereas the C14 homolog achieved its highest value in groups F and H.

The control egg yolk had a tendentially lower percentage of monounsaturated acids (41.7%) than the others in this group. Experiment 2 showed also a significant increase in the amount of palmitoleic acid (C16:1). The studies conducted by Chamruspollert and Sell found that the addition of 5% CLA to the feed resulted in a reduction in the level of monounsaturated fatty acids compared to control eggs [46]. Other researchers such as Aydin [47,48] and Raes et al. have reported similar observations [49]. Additionally, a decrease in the level of C17 and C18 (oleic acid) acids was detected.

In addition, a decrease in the content of C18:3 *n*-3 acid (α-linolenic acid) and C18:2 *n*-6 (linoleic acid) was observed with an increase in the amount of PSO in the feed of laying hens. The experiment also showed that PSO in feed influences the presence of C20 PUFA. This includes C20:2 n-9 acid as well as C20:3 *n*-6 homologs, while the level of C20:4 *n*-6 remains constant (*p* > 0.05).

In other research, it was shown that adding olive leaves to the feed of laying hens did not affect the content of PUFA and monounsaturated fatty acids compared to control eggs [50]. In contrast, the experiment carried out by Kostogrys et al. found the lowest level of polyunsaturated fatty acids in the yolk of hens fed a diet containing 1.5% CLnA [4].

Nutritional intervention in both experiments caused a slight increase in the proportion of saturated acids; however, the observed changes were not statistically significant. It is important to mention that the increase in saturated fatty acids was lower than what other researchers discovered when examining the impact of CLA inclusion in chicken feed on the fatty acid composition of eggs [26,51,52,53,54]. In terms of nutrition, an increase in saturated fatty acids is undesirable due to their atherosclerotic effects [55]. However, Silveirio suggested that a high content of saturated acids may be linked to a slower progression of lipid oxidation in the yolk [52].

### 2.3. Comparative Analysis between Experiments 1 and 2

The eggs obtained in Experiments 1 and 2 did not have significantly different physicochemical properties, except for the color measured using the instrumental method (Table 4). In Experiment 1, eggs obtained from hens fed a diet containing 1% PSO had lower values for ∆E, a (the share of red color), and b (the share of yellow color) compared to eggs obtained from hens fed a diet without PSO. However, adding 1.5% PSO and 2% LO to the feed only increased the L parameter (brightness).

Significant discrepancies were found in the levels of dry matter and ash in egg whites B and F, as well as C and G, when the laying hens were fed 0.5% and 1% PSO in their diet (Table 5). Additionally, the LO concentration in samples F and G reduced the protein content of egg whites regardless of the PSO level.

A lower proportion of 18:2 *n*-6 acid was observed in eggs from hens fed a diet containing 0.5% PSO and LO, and the same relationship was observed for CLA and CLnA contents in D and H eggs (1.5% PSO). When PSO was increased to 1%, the proportion of CLA and CLnA was higher in eggs from hens receiving LO in their feed. Analysis of the effect of PSO on saturated fatty acids showed that the proportions of C14:0 and C16:0 were lower in eggs from Experiment 1. However, statistical analysis did not reveal any significant differences in the percentages of saturated, mono-, and polyunsaturated fatty acids (Table 6).

It should be also mentioned here that adding pomegranate seed oil to laying feed can lead to a higher market price for the resulting product (eggs). Based on the study’s description of the maximum amount (1.5%) of pomegranate seed oil, it is estimated that the cost of laying feed will increase by approximately 80%. For smaller amounts, the increase will be 25% or approximately 50% for supplementation at 0.5% and 1.0%, respectively. As a result, fortified eggs are considered a niche product for specific consumer groups. According to many authors, modern consumers tend to pay a reasonable price to get the health benefits of consuming functional foods [56,57,58]. Sajdakowska et. al. mentioned that the contemporary Polish population is characterized by a greater receptiveness towards novel food products due to the extensive variety of goods accessible in the free market. Moreover, the number of affluent, well-informed consumers has expanded, encompassing individuals who prioritize understanding a product’s nutritional content and its influence on health [59].

## 3. Materials and Methods

### 3.1. Materials

The research material consisted of eggs obtained from two experiments involving Hy-Line Brown laying hens that were 25 weeks old and kept in individual cages with unlimited access to water and feed. Each bird was housed in a cage measuring 30 × 120 × 50 cm, providing a total floor area of 3600 cm^2^. During the pre-experimental phase, which lasted from 17 to 24 weeks of age, the hens were fed a standard commercial diet. The experimental period spanned 12 weeks, from the 25th to the 36th week of age, and was conducted under standard climatic conditions (19 ± 2 °C; 50–65% relative humidity; 14 L:10 D light program). Ad libitum water and feed were provided. We utilized meticulously balanced feed mixtures, comprising energy, protein, and essential minerals, to ensure optimal results. This approach seeks to prevent overconsumption of feed due to compensatory mechanisms triggered by nutrient deficiencies. By carefully formulating feed compositions, we aimed to achieve a harmonious balance between hen welfare and productivity, thereby preventing reduced productivity due to overfeeding.

In both experiments, hens were randomly divided into four groups, each containing an equal number of 24 hens, and fed feed mixtures with varying proportions of linseed oil and pomegranate seed oil (Table 7).

### 3.2. Methods

#### 3.2.1. Sampling Procedure

Twenty eggs randomly selected from each group were collected and broken. All physical parameters (e.g., weight, shape index) were tested on each egg separately. Next, the average laboratory samples were prepared. Yolks were separated from the albumen and the shell. The physicochemical properties of the raw eggs and the chemical composition of the eggs were analyzed. All chemical analyses were performed in triplicate.

#### 3.2.2. Physicochemical Properties of Raw Eggs

The following parameters were determined in raw eggs:(a)Weight: using the Mettler Toledo PB 602-S/FACT balance, with an accuracy of 0.01 g;(b)Egg shape index [60]: the length (long axis) and width (short axis) of the egg were measured using electronic calipers. The ratio of width to length, expressed as a percentage, was the egg shape index;(c)Shell surface (Ps) [61]: according to Paganelli et al., using the following equation:
$$\mathrm{Ps}=4.835\cdot W^{0.662}$$
where W—weight of the egg, g
(d)Haugh’s units (Hu) [62]: according to Williams et al., using the equation
$$\mathrm{Hu}=100\cdot 1g\left(h_{d}-1.7\cdot W^{0.37}+7.6\right)$$
where: h*_d_*—height of dense egg white, mm; W—weight of the egg, g.

(e)Albumen index (WI) [63]: the length and width of the dense egg white were measured with a caliper, and the height of the dense egg white was measured with a caliper depth gauge. WI was calculated as the height of dense egg white in mm divided by the arithmetic mean of the length and width of the dense egg white in mm;(f)Yolk index (YI) [63]: yolk length and width were measured with a caliper, and yolk height was measured with a using a caliper depth gauge so as not to damage the vitelline membrane of the of the yolk. WI was calculated as height of the egg yolk in mm divided by the arithmetic mean of the length and width of the egg yolk in mm.(g)Percentage of egg white, yolk, and shell were calculated based on the weight of the individual components of the egg, i.e., egg white, yolk, shell, g, and weight of whole egg with shell, g;(h)The egg albumen and yolk pH values were measured by means of the Elmetron CP-411pH-meter with the OSH 10-00 electrode type;(i)Yolk color [4] was calculated based on a 15-degree LaRoche scale, using the Yolk Color Fan from DSM Nutritional Products.(j)Yolk color was also determined using an instrumental method [64] using a CM-3500d spectrophotometer from Konica Minolta, in the CIE system. The following parameters were determined from the measurement: L*—color brightness (L* = 0 black, L* = 100 white), a*—proportion of green (a* < 0) or red (a* > 0), b*—share of blue (b* < 0) or yellow (b* > 0) color. In order to interpret the color changes of yolks depending on the level of CLnA in the laying hens’ feed, the absolute color differences ΔE were calculated.

#### 3.2.3. Basic Chemical Composition of Raw Eggs

The following parameters were determined in raw eggs:(a)Moisture of yolk and white was established by drying the samples in a conventional oven at 98 °C for 24 h according to AOAC method [65].(b)The total fat content in the egg white and yolk was assessed by the use of CO_2_ supercritical extraction: pump pressure—9000 PSI; cell temperature—100 °C; carbon dioxide flow rate—1.3 L/min; static time—5 min; dynamic time—45 min (TFE2000 analyzer, LECO, Saint Joseph, MI, USA USA) [66].(c)In order to analyse the protein content, the nitrogen amount was analyzed using the the TruSpec N LECO Company Analyzer. The analysis was performed by means of the Dumas method, according to PN-EN ISO 16634-1:2008 [67], where values of N% were multiplied by 6.25 in order to calculate the protein %.(d)The ash content was analysed by ashing the samples using a muffle furnace oven at 525 °C for 12 h [68].

#### 3.2.4. Fatty Acids Analysis

Lipids were extracted from the egg yolk by means of CO2 supercritical extraction (pump pressure—9000 PSI; cell temperature—100 °C; carbon dioxide flow rate—1.3 L/min; static time—5 min; dynamic time—45 min) using FAT Ex-tractor TFE 2000 (Leco, St. Joshep, MO, USA) [66]. After the extraction, lipids were methylated using sodium methylate [69]. In detail, 0.1 mL of extracted fat was placed in the glass test tube of 2 mL capacity, and 0.5 mL of 0.025 M of sodium methylate solution was added. The mixture was heated in a closed tube at 60 °C until the mixture was clear. The analysis of fatty acids was carried out using gas chromatography (Trace GC Ultra, Thermo Electron Corporation, Beverly, MA, USA). For the analysis, a Supelcowax 10 column (dimensions 30 m × 0.25 mm × 0.25 µm) was used. Helium was used as a sample carrier with a flow rate of 5 mL/min. The injector temperature was 220 °C. The temperature of the column was kept for 3 min at 60 °C, then increased at a rate of 7 °C/min up to 200 °C, and then held at this temperature for 20 min. The detector temperature was set to 250 °C and the split flow was 10 mL/min. Peak identification was performed using an external standard (FIM/FAME Supelco). Fatty acid methyl esters were identified by comparing their retention times with authentic standards (Sigma Aldrich, Poznań, Poland) as well as the Punicic Acid Standard (Larodan Fine Chemicals AB, Malmö, Sweden).

#### 3.2.5. Statistical Analysis

Statistical analysis was performed using StatiStica v. 9.0 (Statsoft, Krakow, Poland) software. Mean values and standard deviations were calculated. The results of the evaluation of the chemical compositions of eggs obtained in experiments 1 and 2 were subjected to a one-way analysis of variance, and the significance of differences between the means (with a significance level of *p* ≤ 0.05) was determined using Duncan’s test. The significance of the differences is demonstrated by the *p*-values displayed in Appendix A. The hypothesis of the normality of the distribution of results was verified using the Shapiro–Wilk test and the homogeneity of variance was verified with Lavene’s test. Moreover, in order to compare the results of the groups fed the feed with the same proportion of pomegranate seed oil from both experiments, the statistical analysis involved determining the significance of the differences between them using the Student’s *t*-test (significance level *p* < 0.05; Table 4, Table 5 and Table 6).

## 4. Conclusions

As a result of modifying the feed composition for laying hens, the eggs retained their high nutritional value while also altering the fatty acid profile in a favorable manner. The simultaneous inclusion of 1.5% linseed oil and pomegranate seed oil in the diet resulted in decreased levels of protein, *n*-6, and *n*-3 acids in egg yolks, and an increased amount of CLA.

The presence of conjugated linolenic acid trienes, obtained through nutritional modification of laying hen feed, in fresh eggs allows for their use as a source of these compounds in human diets. These findings suggest the need for further research to confirm the health-promoting properties of eggs enriched with CLnA and CLA.

## Figures and Tables

**Table 1 molecules-29-01242-t001:** Physicochemical properties of raw eggs.

Parameter	Experiment 1				Experiment 2			
	A	B	C	D	E	F	G	H
Weight (g)	61.08 ±3.30	61.68 ±3.56	62.68 ±3.94	60.46 ±3.93	61.01 ±3.21	61.34 ±3.55	60.83 ±2.43	63.14 ±3.67
Shape index (%)	79.05 ±2.87	78.6 ±2.23	77.57 ±2.62	78.04 ±1.50	79.46 ±2.85	78.40 ±1.58	79.18 ±2.66	79.25 ±2.21
Haugh unit	45.57 ±2.23	45.85 ±2.64	43.56 ±3.81	43.89 ±2.24	45.90 ±2.33	44.36 ±2.21	44.63 ±1.67	44.07 ±1.70
Egg yolk pH	6.01 ±0.09	6.03 ±0.16	6.12 ±0.12	6.16 ±0.29	6.03 ±0.16	6.06 ±0.06	6.06 ±0.09	6.14 ±0.15
Egg white pH	7.14 ^ab^ ± 0.22	7.04 ^a^ ±0.25	7.23 ^ab^ ± 0.24	7.34 ^b^ ±0.16	7.15 ±0.18	7.31 ±0.19	7.37 ±0.12	7.44 ±0.14
Egg yolk index	0.42 ±0.03	0.43 ±0.02	0.42 ±0.02	0.44 ±0.02	0.43 ^B^ ±0.04	0.43 ^B^ ±0.03	0.41 ^B^ ±0.02	0.36 ^A^ ±0.02
Egg white index	0.09 ^b^ ±0.02	0.09 ^b^ ±0.02	0.08 ^ab^ ± 0.01	0.07 ^a^ ±0.01	0.09 ^B^ ±0.02	0.08 ^A^ ±0.01	0.07 ^A^ ±0.01	0.08 ^A^ ±0.01
Shell share (%)	12.83 ±0.95	12.49 ±1.17	11.52 ±0.01	12.79 ±0.01	12.71 ±0.63	12.61 ±1.07	12.35 ±0.57	12.08 ±0.84
Egg white share (%)	61.78 ±1.72	61.78 ±1.28	62.44 ±2.10	61.59 ±1.28	62.31 ±1.97	62.78 ±1.42	62.62 ±1.47	61.96 ±2.65
Egg yolk share (%)	25.37 ±1.37	25.72 ±1.79	25.63 ±1.34	25.6 ±1.18	24.96 ±1.64	24.59 ±1.02	25.02 ±1.82	25.95 ±2.07
Egg yolk color (LaRoche scale)	11.00 ±0.00	12.00 ±0.00	12.00 ±1.73	11.00 ±0.00	11.66 ±0.57	12.66 ±0.57	10.33 ±1.52	11.33 ±0.57
∆E ^1^	-	5.35 ^c^ ±3.09	4.29 ^b^ ±1.89	2.75 ^a^ ±1.89	-	7.52 ^A^ ±2.41	6.61 ^A^ ±2.65	11.22 ^B^ ±4.80
L* ^2^	41.88 ±3.07	44.28 ±4.37	40.73 ±3.67	42.20 ±2.20	44.35 ^C^ ± 4.06	39.55 ^B^ ± 3.49	39.84 ^B^ ± 3.02	36.73 ^A^ ±7.46
a* ^3^	11.36 ±41.15	11.56 ±0.89	11.79 ±1.58	10.45 ±0.94	12.73 ^C^ ± 1.28	12.80 ^D^ ± 1.23	9.55 ^A^ ±1.07	10.32 ^B^ ±1.19
b* ^4^	27.88 ±2.81	29.22 ±3.63	26.78 ±2.25	28.38 ±1.68	31.49 ±4.45	27.32 ±3.17	28.77 ±1.82	28.00 ±4.65

^a,b,c^—values marked with different letters are significantly different *p* ≤ 0.05 (Experiment 1). ^A,B,C,D^—values marked with different letters are significantly different *p* ≤ 0.05 (Experiment 2). ^1^—color difference. ^2^—lightness. ^3^—red/green value. ^4^—blue/yellow value.

**Table 2 molecules-29-01242-t002:** Basic chemical composition of eggs.

Parameter		Experiment 1				Experiment 2			
		A	B	C	D	E	F	G	H
Dry matter (g/100 g)	Y ^1^	47.75 ^bc^ ±0.00	47.27 ^a^ ±0.25	47.48 ^ab^ ±0.15	48.08 ^c^ ±0.12	46.11 ^A^ ±0.30	46.85 ^AB^ ±0.38	47.78 ^B^ ±0.01	46.84 ^AB^ ±0.64
	W	13.44 ^c^ ±0.02	13.22 ^b^ ±0.03	12.90 ^a^ ±0.01	12.96 ^a^ ±0.04	13.64 ^D^ ±0.14	12.74 ^A^ ±0.00	13.38 ^B^ ±0.04	13.10 ^B^ ±0.05
Protein (g/100 g of d.m.)	Y	34.10 ±0.26	33.78 ±0.29	33.96 ±0.04	33.8 ±0.27	34.90 ^B^ ±0.92	34.63 ^B^ ±0.43	34.72 ^B^ ±0.07	32.78 ^A^ ±0.98
	W	84.3 ^a^ ±0.26	84.3 ^a^ ±0.35	86.78 ^b^ ±0.21	87.92 ^c^ ±0.15	82.79 ^A^ ±0.35	88.10 ^B^ ±0.14	88.39 ^BC^ ±0.09	88.60 ^C^ ±0.03
Fat (g/100 g of d.m.)	Y	50.42 ±0.31	50.54 ±1.83	49.18 ±0.62	52.99 ±3.21	52.24 ±0.97	50.91 ±0.33	50.82 ±1.73	49.92 ±2.69
Ash (g/100 g of d.m.)	Y	4.73 ±0.20	4.19 ±0.12	4.57 ±0.19	4.66 ±0.06	5.37 ^B^ ±0.08	4.74 ^A^ ±0.28	4.71 ^A^ ±0.04	4.78 ^A^ ±0.07
	W	6.08 ±0.02	5.60 ±0.06	6.05 ±0.01	6.62 ±0.18	6.14 ±0.06	6.10 ±0.05	6.24 ±0.06	6.14 ±0.12

^1^ Y—egg yolk, W—egg white. ^a,b,c^—values marked with different letters are significantly different *p* ≤ 0.05 (Experiment 1). ^A,B,C,D^—values marked with different letters are significantly different *p* ≤ 0.05 (Experiment 2).

**Table 3 molecules-29-01242-t003:** Fatty acids profile of egg yolks (%).

Acid	Experiment 1				Experiment 2			
	A	B	C	D	E	F	G	H
Saturated								
C14:0 Tetradecanoic (myristic acid)	0.66 ^a^ ±0.01	0.64 ^a^ ±0.03	0.72 ^b^ ±0.00	0.65 ^a^ ±0.01	0.67 ^AB^ ± 0.00	0.81 ^C^ ±0.04	0.67 ^AB^ ±0.00	0.82 ^C^ ±0.03
C15:0 Pentadecanoic (pentadecylic acid)	0.08 ±0.00	0.09 ±0.01	0.08 ±0.00	0.10 ±0.00	0.10 ±0.00	0.10 ±0.00	0.09 ±0.00	0.11 ±0.01
C16:0 Hexadecanoic (palmitic acid)	20.41 ^b^ ±0.08	20.44 ^b^ ±0.14	19.71 ^a^ ±0.04	20.16 ^ab^ ±0.31	20.92 ±0.30	20.86 ±0.20	21.25 ±0.12	20.96 ±0.46
C17:0 Heptadecanoic (margaric acid)	0.31 ±0.02	0.31 ±0.05	0.36 ±0.02	0.36 ±0.04	0.29 ±0.05	0.36 ±0.00	0.37 ±0.00	0.35 ±0.00
C18:0 Octadecanoic (stearic acid)	9.42 ^a^ ±0.09	9.85 ^ab^ ±0.25	10.54 ^b^ ±0.30	9.89 ^ab^ ±0.55	8.09 ^A^ ±1.00	8.82 ^AB^ ± 0.27	9.32 ^B^ ±0.42	9.84 ^B^ ±0.00
C22:0 Docosanoic (behenic acid)	0.00 ±0.00	0.00 ±0.00	0.00 ±0.00	0.00 ±0.00	0.00 ±0.00	0.00 ±0.00	0.00 ±0.00	0.04 ±0.06
Monounsaturated								
C14:1 9-tetradecenoic (myristoleic acid)	0.12 ±0.00	0.12 ±0.01	0.12 ±0.00	0.11 ±0.00	0.12 ^A^ ±0.01	0.14 ^B^ ±0.01	0.10 ^A^ ±0.00	0.13 ^A^ ±0.01
C16:1 *Trans*-3-hexadecenoic	2.17 ^b^ ±0.00	1.91 ^a^ ±0.13	1.80 ^a^ ±0.07	1.70 ^a^ ±0.07	2.64 ^C^ ±0.05	2.12 ^AB^ ±0.02	2.13 ^AB^ ±0.05	1.89 ^B^ ±0.06
C16:1 9-cis-hexadecenoic (palimitoleic acid)	5.91 ^a^ ±0.14	5.53 ^b^ ±0.03	5.18 ^b^ ±0.13	5.24 ^b^ ±0.36	4.41 ^A^ ±0.09	5.63 ^B^ ±0.08	4.74 ^BC^ ±0.11	4.95 ^BC^ ±0.43
C17:1 10-heptadecenoic acid	0.36 ±0.01	0.33 ±0.00	0.32 ±0.00	0.3 ±0.09	0.33 ^B^ ±0.00	0.32 ^AB^ ±0.00	0.29 ^AB^ ±0.00	0.23 ^A^ ±0.01
C18:1 *cis*-9-octadecenoic (oleic acid)	40.88 ^b^ ±0.02	38.53 ^c^ ± 0.38	35.96 ^a^ ±0.12	34.33 ^a^ ±1.16	42.51 ^C^ ± 0.28	39.2 ^B^ ±0.36	39.34 ^B^ ±0.55	36.00 ^A^ ± 0.53
Polyunsaturated acids								
C16:2 Hexadecadienoic acid	0.08 ±0.02	0.06 ±0.00	0.07 ±0.00	0.06 ±0.00	0.08 ^C^ ±0.00	0.08 ^BC^ ± 0.01	0.06 ^AB^ ±0.00	0.05 ^A^ ±0.00
C18:2 *n*-6 *cis,cis*-9,12-octadecadienoic (linoleic acid)	13.42 ^a^ ±0.01	13.33 ^a^ ± 0.01	13.36 ^a^ ±0.01	13.75 ^b^ ±0.19	15.46 ^C^ ±0.47	14.11 ^B^ ± 0.04	13.30 ^AB^ ±0.03	13.81 ^A^ ± 0.15
C18:2 Conjugated linoleic acids—CLA	0.26 ^a^ ±0.02	1.92 ^b^ ±0.08	3.39 ^c^ ±0.09	3.96 ^d^ ±0.14	0.00 ^A^ ±0.00	2.25 ^B^ ±0.05	3.23 ^C^ ±0.04	4.33 ^D^ ±0.13
C18:3 *n*-3 *Cis,cis,cis*-9,12,15-octadecatrienoic (α-linolenic acid)	5.18 ^ab^ ±0.12	4.89 ^a^ ±0.07	5.38 ^b^ ±0.06	5.08 ^ab^ ±0.18	3.54 ^A^ ±0.34	2.80 ^B^ ±0.09	2.17 ^CD^ ±0.05	2.09 ^D^ ±0.12
C18:3 Conjugated linolenic acid— CLnA	0.00 ^a^ ±0.00	0.85 ^b^ ±0.06	1.58 ^c^ ±0.07	2.63 ^d^ ±0.39	0.00 ^A^ ±0.00	0.92 ^B^ ±0.04	1.41 ^C^ ±0.09	2.53^D^ ±0.32
C20:2 Eicosadienoic	0.06 ±0.00	0.07 ±0.02	0.08 ±0.00	0.07 ±0.00	0.08 ^A^ ±0.00	0.09 ^A^ ±0.00	0.07 ^B^ ±0.00	0.10 ^C^ ±0.00
C20:3 *n*-6 *Cis,cis,cis*-8,11,14-eicosatrienoic dihomo-γ-linolenic acid	0.06 ±0.00	0.05 ±0.02	0.07 ±0.00	0.06 ±0.00	0.05 ^A^ ±0.00	0.07 ^B^ ±0.00	0.06 ^B^ ±0.00	0.09 ^C^ ±0.00
C20:4 *n*-6 5,8,11,14-*all-cis*-eicosatetraenoic (arachidonic acid)	0.22 ^a^ ±0.02	0.34 ^b^ ±0.01	0.34 ^b^ ±0.00	0.38 ^b^ ±0.02	0.24 ±0.04	0.42 ±0.04	0.42 ±0.14	0.41 ±0.04
C22:6 *n*-3 Docosahexaenoic—DHA (cervonic acid)	0.11 ^a^ ±0.02	0.17 ^b^ ±0.00	0.17 ^b^ ±0.01	0.18 ^b^ ±0.02	0.06 ±0.00	0.12 ±0.02	0.11 ±0.06	0.08 ±0.02
Saturated fatty acids—SFA (%)	30.07 ±0.62	30.88 ±0.04	31.47 ±0.40	31.43 ±0.29	31.17 ±0.26	30.96 ±0.12	31.71 ±0.30	32.09 ±0.41
Monounsaturated fatty acids— MUFA (%)	50.02 ±0.22	49.45 ±0.14	46.44 ±0.55	43.39 ±0.31	41.70 ±0.77	47.41 ±0.32	46.61 ±0.62	43.27 ±0.12
Polyunsaturated fatty acids— PUFA (%)	19.90 ±0.85	19.66 ±0.09	22.22 ±0.33	25.17 ±0.02	27.13 ±0.50	21.62 ± 0.20	21.67 ±0.31	24.59 ±0.46

^a,b,c,d^—values marked with different letters are significantly different *p* ≤ 0.05 (Experiment 1). ^A,B,C,D^—values marked with different letters are significantly different *p* ≤ 0.05 (Experiment 2).

**Table 4 molecules-29-01242-t004:** Comparison of basic physicochemical properties of eggs obtained in Experiments 1 and 2—*p*-values (Student’s *t*-test; significance level *p* < 0.05).

Parameter	Type of Eggs			
	A vs. E	B vs. F	C vs. G	D vs. H
Weight	0.963	0.832	0.221	0.132
Shape index	0.751	0.820	0.189	0.169
Haugh unit	0.750	0.189	0.425	0.844
Egg yolk pH	0.644	0.600	0.240	0.852
Egg white pH	0.905	**0.014**	0.125	0.164
Egg yolk index	0.493	0.967	0.396	**0.000**
Egg white index	0.626	0.060	0.962	0.060
Shell share	0.749	0.807	0.311	0.077
Egg white share	0.533	0.114	0.830	0.695
Egg yolk share	0.552	0.101	0.405	0.652
Egg yolk color (LaRoche scale)	0.116	0.373	0.279	0.373
∆E	-	0.098	**0.037**	**0.000**
L*	0.141	**0.015**	0.559	**0.039**
a*	**0.021**	**0.018**	**0.001**	0.792
b*	**0.044**	0.227	0.043	0.810

Statistically significant *p*-values were designated in bold.

**Table 5 molecules-29-01242-t005:** Comparison of the basic chemical composition of eggs obtained in Experiments 1 and 2—*p*-values (Student’s *t*-test; significance level *p* <0.05).

Parameter		Type of Eggs			
		A vs. E	B vs. F	C vs. G	D vs. H
Dry matter	Y ^1^	**0.016**	0.320	0.102	0.113
	W	0.177	**0.001**	**0.004**	0.079
Protein	Y	0.225	**0.050**	**0.009**	0.159
	W	**0.004**	**0.000**	**0.000**	**0.001**
Fat	Y	0.129	0.805	0.333	0.409
Ash	Y	0.057	0.127	0.441	0.204
	W	0.388	**0.015**	0.058	0.088

^1^ Y—egg yolk. W—egg white. Statistically significant *p*-values were designated in bold.

**Table 6 molecules-29-01242-t006:** Comparison of fatty acid profiles of eggs obtained in Experiments 1 and 2—*p*-values (Student’s *t*-test; significance level *p* < 0.05).

Parameter	Type of Eggs			
	A vs. E	B vs. F	C vs. G	D vs. H
**Polyunsaturated acids**				
C16:2	**0.037**	0.807	**0.000**	0.095
C18:2 *n*-6	**0.049**	**0.002**	0.451	0.055
C18:2	**0.001**	**0.000**	**0.002**	**0.014**
C18:3 *n*-3	**0.031**	**0.002**	**0.001**	**0.001**
C18:3	**0.011**	**0.001**	**0.019**	0.056
C20:2 n-9	0.292	**0.037**	0.770	0.105
C20:3 *n*-6	**0.000**	0.292	0.591	**0.000**
C20:4 *n*-6	0.066	**0.028**	0.504	0.166
C22:6 *n*-3	**0.014**	0.683	0.316	**0.042**
Remaining acids: C18:2, C18:3, CLA	**0.013**	**0.012**	0.072	0.051
**Monounsaturated acids**				
C14:1	0.698	0.311	0.311	0.698
C16:1 (3t-hexadecanoic)	**0.005**	0.116	0.172	0.293
C16:1 (9-hexadecanoic)	0.087	0.138	**0.010**	0.546
C17:1	0.667	0.057	**0.029**	**0.044**
C18:1	**0.010**	**0.023**	0.233	0.810
**Saturated acids**				
C14:0	0.183	**0.041**	0.360	0.059
C15:0	**0.000**	**0.000**	0.698	0.154
C16:0	0.139	0.101	**0.028**	0.063
C17:0	0.293	0.116	0.248	0.422
C18:0	0.156	0.100	0.269	0.082
C22:0	**0.000**	**0.000**	**0.000**	0.422
The share of polyunsaturated acids	0.887	0.978	0.948	0.984
The share of monounsaturated acids	0.488	0.982	0.998	0.993
The share of saturated acids	0.934	0.978	0.950	0.999

Statistically significant *p*-values were designated in bold.

**Table 7 molecules-29-01242-t007:** The content of plant oils in experimental groups.

Group ID	Plant Oil Content, %		
	Rapeseed	Linseed	Pomegranate
Experiment 1			
A	2.5	1.5	0.0
B	2.0	1.5	0.5
C	1.5	1.5	1.0
D	1.0	1.5	1.5
Experiment 2			
E	4.0	0.0	0.0
F	3.5	0.0	0.5
G	3.0	0.0	1.0
H	2.5	0.0	1.5

The details on feed composition as well as fatty acid profile of oils used in all experiments are presented in Appendix A.

## Data Availability

The data are included in this article and Appendix A.

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
