# Peer review of "Design of Physicochemical Properties of Eggs as a Result of Modification of the Fat Fraction of Laying Feed"

_molecules, 2024, doi:10.3390/molecules29061242_

Round 1

Reviewer 1 Report

Comments and Suggestions for Authors

The article was about the modification of fat fraction of laying feed on the properties of eggs. The study was meaningful. The article was well written. The results were discussed properly. So I recommend acceptance after revising the following suggestions.

Line 32: Delete the . before [3-5].

Line 123: It would be clearer to give the experimental groups by table

Line 194: Give the specific method and parameters set of CO2 supercritical extraction 

Reviewer 2 Report

Comments and Suggestions for Authors

The manuscript "Design of physicochemical properties of eggs as a result of modification of the fat fraction of laying feed" reports the results of a research on the use of pomegranate seed and rapeseed as the main source of linoleic and linolenic conjugated fatty acids (CLA and CLnA) in laying hen diets. For this purpose, the quality of the eggs was assessed using commercial parameters and nutritional properties.

Specific comments:

Line 120 use lower case.

Line 201 is 0.25 mm?

Line 217 Levene’s test

Line 229 Delete “experiments”

Table 1 Use L*, a*, b* as indicated in M& M and not L, a and b. Why do the Authors use Luminance and not lightness or brightness, which are more commonly used?

Line 247 The DE value depends on the L*, a* and b* values, but in experiment 1 there are no significant differences in these parameters. Your sentence seems to emphasise the opposite.

Line 268-270 About the C16 FA content. The value in group D is not statistically different from the other groups. I suggest to correct the sentence.

Line 273 p>0.05

Line 274 and 275 trans-3-hexadecenoic acid, please correct

Line 276 and 277 check the highest values of FA observed in C and D groups

Lines 296 and 297: see comment on Table 1. The sentence is not clear, the a* value (red index) is significantly different between groups, and the b* value (yellow index; blue=negative and yellow=positive) doesn't change significantly.

Line 300 -313 I suggest checking the commentary on the results, which are incorrect with respect to the values given in the table. In lines 307 and 308, do the authors refer to egg yolk protein or egg white protein?

Line 314 Insert reference to Table 3.

Line 320 In group F? check

Line 321 When the results are not significant I suggest to use a sentence with “tendentially”

Line 327 C17:1 but also C18:1 (oleic acid)

Table 3.  C16:1 9-cis-hexadecenoic. The value 5.91 has ab as exponent letters. I think the letter a is enough.  

Table 3A There is not luminance in Table, so change the footnotes.
